# Severity predictors of COVID-19 in SARS-CoV-2 variant, delta and omicron period; single center study

Fumihiro Ogawa[1]*, Yasufumi Oi[1], Hiroshi Honzawa[1], Naho Misawa[1], Tomoaki Takeda[1], Yushi Kikuchi[1], Ryosuke Fukui[1], Katsushi Tanaka[2], Daiki Kano[2], Hideaki Kato[2], Takeru Abe[1], Ichiro Takeuchi[1]

1 Department of Emergency Medicine, Yokohama City University, School of Medicine, Yokohama, Kanagawa, Japan, 2 Infection Prevention and Control Department, Yokohama City University Hospital, Yokohama, Kanagawa, Japan

* fumihiro@yokohama-cu.ac.jp

**Data Availability Statement:** The data used in this paper were acquired from https://doi.org/10.6084/m9.figshare.20652678.v2.

## Abstract

### Background

The outcomes of coronavirus disease 2019 (COVID-19) treatment have improved due to vaccination and the establishment of better treatment regimens. However, the emergence of variants of the severe acute respiratory syndrome coronavirus 2 (SARS-CoV-2), which causes COVID-19, and the corresponding changes in the characteristics of the disease present new challenges in patient management. This study aimed to analyze predictors of COVID-19 severity caused by the delta and omicron variants of SARS-CoV-2.

### Methods

We retrospectively analyzed the data of patients who were admitted for COVID-19 at Yokohama City University Hospital from August 2021 to March 2022.

### Results

A total of 141 patients were included in this study. Of these, 91 had moderate COVID-19, whereas 50 had severe COVID-19. There were significant differences in sex, vaccination status, dyspnea, sore throat symptoms, and body mass index (BMI) (p <0.0001, p <0.001, p <0.001, p = 0.02, p< 0.0001, respectively) between the moderate and severe COVID-19 groups. Regarding comorbidities, smoking habit and renal dysfunction were significantly different between the two groups (p = 0.007 and p = 0.01, respectively). Regarding laboratory data, only LDH level on the first day of hospitalization was significantly different between the two groups (*p*<0.001). Multiple logistic regression analysis revealed that time from the onset of COVID-19 to hospitalization, BMI, smoking habit, and LDH level were significantly different between the two groups (p<0.03, p = 0.039, p = 0.008, p<0.001, respectively). The cut-off value for the time from onset of COVID-19 to hospitalization was four days (sensitivity, 0.73; specificity, 0.70).

**Funding:** The authors received no specific funding for this work.

**Competing interests:** The authors received no specific competing interests for this work.

## Conclusions

Time from the onset of COVID-19 to hospitalization is the most important factor in the prevention of the aggravation of COVID-19 caused by the delta and omicron SARS-CoV-2 variants. Appropriate medical management within four days after the onset of COVID-19 is essential for preventing the progression of COVID-19, especially in patients with smoking habits.

## Introduction

The coronavirus disease 2019 (COVID-19) pandemic has caused a significant increase in hospitalizations for pneumonia with multiorgan disease. COVID-19 is caused by the novel severe acute respiratory syndrome coronavirus 2 (SARS-CoV-2). Vaccines and treatment methods against SARS-CoV-2 are increasingly being developed worldwide. Although the number of patients with COVID-19 is still high, these measures have led to the reduction of disease severity and improvement of prognosis after the onset of symptoms. However, despite tremendous efforts by scientists, researchers, and health practitioners to combat the COVID-19 pandemic, the emergence of new variants of SARS-CoV-2 present new challenges in patient management. As the development of vaccines and treatments progresses and patient outcomes improve, SARS-CoV-2 repeatedly mutates and tries to survive these improved therapies. The World Health Organization and Centers for Disease Control and Prevention have identified the alpha, beta, gamma, delta, epsilon, eta, lota, kappa, mu, zeta, and omicron variants of SARS-CoV-2 [1, 2]. The recently identified delta and omicron variants are both highly infectious and are different from the previous strains in terms of infectivity, severity, and clinical symptoms. In addition, it is unknown what an optimal timing for hospitalization is to prevent exacerbation. Thus, retrospective analysis of the characteristics and aggravation markers of patients newly infected with the delta and omicron variants of SARS-CoV-2 is essential and will lay the groundwork for the management of the emergence of new mutant strains of SARS-CoV-2. Therefore, this study aimed to analyze and describe the characteristics, clinical features, and outcomes of COVID-19 caused by the delta and omicron variants of SARS-CoV-2 according to disease severity. We also identified an optimal cut-off point between onset and hospitalization for lower severity.

## Patients and methods

This was a retrospective study conducted using the data of patients with COVID-19 (except for outpatients with mild COVID-19) who underwent standard treatment, including intensive care, at Yokohama City University Hospital from June 2021 to March 2022. The clinical and biological features of the patients were analyzed. During the observation period, June 2021 to December 2021 and January 2022 to March 2022 were marked by a surge in the number of infections caused by the delta and omicron variants of SARS-CoV-2, respectively. Thus, we defined June 2021 to December 2021 as the 'delta period' and January 2022 to March 2022 as the 'omicron period.' If a patient with mild COVID-19 had a severe risk factor, such as old age, chronic kidney disease that requires hemodialysis, or severe immunosuppression, we judged the patient's condition and COVID-19 severity and decided on hospitalization. The characteristics, risk factors, morbidity, and mortality outcomes of the patients were analyzed as well. Information on comorbidities were obtained for each patient, and outcome data were obtained during follow-up in our hospital.

This study was approved by the Ethics Committee of the Yokohama City University School of Medicine. Written informed consent was obtained from each patient or their family members before treatment.

## Classification of coronavirus disease severity

Disease severity was categorized into the mild-moderate stage (moderate group) and the severe-critical stage (severe group) based on previously published guidelines on the diagnosis and treatment of COVID-19 [3]. Mild cases were defined as patients with no symptoms, no need for oxygen, and no sign of pneumonia on computed tomography (CT) scans. Moderate cases were defined as patients with mild respiratory symptoms, radiological evidence of pneumonia, and oxygen saturation ($SpO_2$) >93% and <96%. Severe cases were defined as patients with $SpO_2$ ≤92% and requiring oxygen support. Critical cases were defined as patients that required heart–lung machine or extracorporeal membrane oxygenation (ECMO) support for acute respiratory distress syndrome (ARDS). Severe and critical cases were combined into the same group in this study. We did not consider CT score [4, 5] in the classification of COVID-19 severity.

## Sample collection

The blood samples of the patients were collected at admission depending on the clinical conditions of the patients with severe COVID-19 who needed oxygen and those in critical conditions who needed intubation management. Table 1 shows the severity classification and treatment strategies for the patients with critical COVID-19.

On admission to our hospital, COVID-19 was diagnosed using a positive reverse transcriptase–polymerase chain reaction (RT-PCR) assay for SARS-CoV-2. The RT-PCR assay was performed using respiratory tract and laryngeal swab samples, which were sent to a designated diagnostic laboratory. Standard procedures for sample collection were used to ensure that all the samples were treated rapidly and equally.

## Data collection

Patients were followed up until hospital discharge or death. The patient information collected included demographic characteristics, pre-existing comorbidities on the date of hospitalization, and laboratory test results. The laboratory tests included measurement of several hemostatic biomarkers, such as white blood cell count (WBC) and D-dimer, C-reactive protein (CRP), aspartate aminotransferase (AST), alanine aminotransferase (ALT), lactate dehydrogenase (LDH), blood urea nitrogen (BUN), creatinine (Cre), total bilirubin (T.bil), and interleukin-6 (IL-6) levels, which have been reported to be severity markers of COVID-19. These biomarker levels measured from the day of admission to the day of discharge depending on the patient's condition.

We recorded the clinical interventions administered during the observation period, including the use of antibiotics, antiviral agents, systemic corticosteroids, vasoactive medications, venous thromboembolism prophylaxis, antiplatelet or anticoagulation treatment, renal replacement therapy, high-flow oxygen therapy, and mechanical ventilation (invasive and noninvasive).

## Statistical analysis

Patients were divided into two groups for comparison: the moderate group and the severe group. For each group, medians (interquartile ranges) and frequencies (%) were reported for continuous and categorical variables, respectively. We used the Mann-Whitney U test for

**Table 1. Severity classification criteria and therapeutic strategy for critical COVID-19.**

| | | |
|---|---|---|
| • Criteria for mild, severe or critical COVID-19 | | |
| 1) Moderate or severe: Oxygen demand or with risk factors: age, chronic renal failure, severe obesity, etc. 2) Critical: SpO$_2$ ≤92% at 10L/min. oxygen via a reservoir mask | | |
| 3) Critical: Shortness of breath with respiratory rate of >30/min. | | |
| 4) Critical: Severe dyspnea due to COVID-19 pneumonia • Therapeutic strategy for moderate or severe COVID-19 1) Moderate: Neutralizing antibody 2) Severe: antiviral therapy: Remdesivir 5–10 days 3) Severe: Systemic steroid therapy: Dexamethasone 5–10 days Antibiotics: depend on patients' comorbidities for CAP | | |
| • Therapeutic strategy for critical COVID-19 | | |
| 1)Mechanical ventilator | mode | pressure control |
| (primary setting) | PEEP | 10–15 mmH$_2$O |
| | Driving Pressure | 20–25 mmH$_2$O |
| | Respiratory Rate | 12-16/min. |
| (optional therapies) | Self-prone position | |
| | Pone position | |
| | ECMO | |
| 2) Antiviral therapy | Remdesivir | 5 or 10 days |
| 3) Systemic steroid therapy | Dexamethasone | 10 days |
| 4) Anticoagulant therapy | UFH with therapeutic dose according to APTT (1.5–2 times as normal) | |
| 5) Protection for DVT | Intermittent air compression and elastic stocking | |
| 6) Antibiotics | for CAP or secondary bacterial or fungus infection | |
| 7) Rehabilitation | early intervention by NS, PT and OT | |
| 8) Nutrition | early intervention via tube feeding or TPN | |
| 9) Supportive therapy | sedation, catecholamine support etc. via central venous catheter | |

CAP: community associated pneumonia; PEEP: positive end-expiratory pressure; PS: pressure support; ECMO: Extracorporeal membrane Oxygenation; UFH: unfractionated heparin; APTT: activated partial thromboplastin time; NS: nurse; PT: physical therapist; OT: occupational therapist; TPN: total parenteral nutrition.

analysis of continuous variables and Fisher's exact test for the evaluation of categorical variables. Patient characteristics, time from the onset of COVID-19 or acquisition of a positive PCR result to hospitalization, results of standard blood tests, and physical condition were analyzed. In addition, repeated measures analyses of variance were used to evaluate group and time differences, as well as their interactions, for WBC and D-dimer, CRP, AST, ALT, LDH, BUN, Cre, T.bil, and IL-6 levels. We utilized a multivariable logistic regression model to identify the relationship between COVID-19 severity and patient characteristics, including primary symptoms, comorbidities, and laboratory data. In addition, to determine an optimal cut-off value of the interval between onset and admission for different periods of COVID-19 variants, the receiver operating curve, ROC, analysis was performed. The area under the curve, AUC, sensitivity, specificity, and their 95% confidence interval were calculated. Two-sided P values < 0.05 were considered statistically significant. All statistics analyses were conducted using JMP Pro Version 15 (SAS Institute Inc., Cary, NC, USA) and IBM SPSS Statistics for Windows, Version 25.0 (Armonk, NY: IBM Corp).

## Results

We analyzed 141 patients with COVID-19 (median age, 58.0 [IQR, 56–62] years) who required hospitalization and were admitted to our hospital during the study period. Baseline

**Table 2. Patients' characteristics.**

| | | All Cases (n = 141) |
|---|---|---|
| Sex (Male, %) | | 99 (70.2%) |
| Age (y.o., Median ±SD, range) | | 58±16.1 (18–93) |
| Period from onset to hospitalization (days, Median ±SD, range) | | 5±4.2 (0–22) |
| Period from onset to PCR positive (days, Median ±SD, range) | | 1±2.3 (0–13) |
| Symptom (cases, %) | | |
| | fever | 126 (89.4) |
| | dyspnea | 79 (56.0) |
| | cough | 54 (38.3) |
| | fatigue | 48 (34.0) |
| | sore throat | 10 7.1) |
| | consciousness disorder | 1 (0.7) |
| | headache | 4 (2.8) |
| Height (cm, median ±SD, range) | | 165±10.1 (123–189) |
| Weight (kg, median ±SD, range) | | 65±17.1 (34–121) |
| BMI (median ±SD, range) | | 24±5.0 (14–46) |
| smoking habit (cases, %) | | 77 (54.6) |
| vaccination (yes, %) | | 55 (39.0) |
| comorbidities (cases, %) | | |
| | respiratory disease | 20 (14.2) |
| | cardiovascular disease | 29 (20.6) |
| | renal disease | 40 (28.4) |
| | continuous hemodialysis | 30 (21.3) |
| | diabetes | 38 (27.0) |
| | hypertension | 61 (43.3) |
| | hyperlipidemia | 25 (17.7) |
| | collagen diseases | 5 (3.5) |
| | with malignant tumor | 9 (6.4) |
| | pregnancy | 2 (1.4) |
| | immunosuppression drugs | 7 (5.0) |

y.o.: year-old, SD: Standard Deviation, PCR: Polymerase Chain Reaction

BMI: Body Mass Index

demographic characteristics, sex, age, time from onset of COVID-19 to hospitalization, primary symptoms, physical characteristics, smoking habits, vaccination status, and comorbidities of the patients are reported in Table 2. Characteristics of patients admitted during the delta and omicron periods are summarized in S1 Table. The data showed significant differences in patients' age, time from onset to hospitalization, dyspnea as a primary symptom, weight, body mass index (BMI), cardiac disease, renal dysfunction, continuous hemodialysis, and hypertension as comorbidities between the two periods ($p < 0.001$, $p < 0.001$, p = 0.03, p = 0.027, p = 0.024, $p < 0.001$, p = 0.0003, p = 0.01, and p = 0.003, respectively). The major difference between the delta and omicron periods was believed to be attributable the effect of vaccination on the public.

## Classification of coronavirus disease severity

Of the 141 patients analyzed in this study, 91 had moderate COVID-19 and did not need intensive care, whereas 50 had severe COVID-19 and needed mechanical ventilation. The

characteristics, primary symptoms, comorbidities, and primary laboratory data of the patients classified according to COVID-19 severity are shown in Table 3.

There was no significant difference in age between the moderate group (59±17.5 years; range, 18–93 years) and the severe group (57±13.3 years; range, 30–93 years) (p = 0.312). Male patients were more likely to progress to severe COVID-19 than were female patients (p<0.001). Time from the onset of COVID-19 symptoms to the day of hospitalization was significantly longer in the severe group than in the moderate group (moderate group: 3±3.6 days; range, 0–13 days vs. severe group: 8±4.2 days; range, 0–22 days) (p<0.001). However, there was no significant difference between the two groups in terms of the time from the acquisition of a positive PCR test result to hospitalization (moderate group: 1±2.3 days; range, 0–13 days vs. severe group: 2±2.3 days, range, 0–10 days) (p = 0.062). Regarding primary symptoms, there was a significant difference in the frequency of dyspnea between the two groups (p<0.001). Regarding physical features, patients in the severe group generally had a greater body weight (59.9±16.0 kg vs. 74.1±16.5 kg) and higher BMI (22.8±10.8 vs. 26.3±8.0) than those in the moderate group (p<0.001 and p<0.001, respectively). Smoking habit was significantly different between the two groups (p = 0.007). No significant frequency of any comorbidity was observed in the severe group. However, renal disease was significantly more frequent in the moderate group than in the severe group (p = 0.016).

Regarding laboratory data on the day of admission, there were significant differences in AST (34.0±40.2 IU/l vs. 50.5±46.9 IU/l), ALT (21.0±30.0 IU/l vs. 50.5±41.3 IU/l), and LDH (300.0±136.7 IU/l vs. 499.5±192.1 IU/l) levels between the severe and moderate groups (p<0.001, p<0.001, and p<0.001, respectively). However, differences in IL-6 (52.3±121.7 pg/ml vs. 31.3±75.3 pg/ml), CRP (5.7±6.8 mg/dl vs. 6.4±7.5 mg/dl), and D-dimer (1.2±13.2 μg/ml vs. 1.1±9.6 μg/ml) levels between the two groups were not statistically significant (p = 0.162, p = 0.211, and p = 0.852, respectively). S1 Fig showed box plot for the relationship between severity of COVID-19 and clinical data represented by continuous variables.

Table 4 shows therapeutic strategies implemented for the treatment of the cases of moderate and severe COVID-19. There was a significant difference between the moderate and severe groups in terms of oxygen demand and intensive care beyond ventilator management. In addition, there were differences in treatment between the two groups in terms of the characteristics of each treatment due to differences in treatment between moderate and severe cases of COVID-19 as indicated by therapeutic strategy in Table 1.

The above-mentioned results regarding the characteristics and clinical features of the moderate and severe groups, as well as the clinical characteristics of the patients admitted during the delta and omicron periods stratified according to disease severity, are outlined in S2 Table. The greatest difference between the delta and omicron periods was the difference in the spread of vaccination. Thus, there were some significant differences in vaccination-associated factors between the two periods.

## Multivariable logistic regression analysis

Multivariable logistic regression analysis was performed to identify the relationship between COVID-19 severity and patient characteristics, including primary symptoms, comorbidities, and laboratory data. The results showed that time from the onset of COVID-19 to hospitalization, BMI, smoking habits, and LDH level were significantly associated with the COVID-19 severity (OR = 1.16, p = 0.026; OR = 1.10, p = 0.039; OR = 3.70, p = 0.008; and OR = 1.01, p< 0.001, respectively) (Table 5). The ROC curve (Fig 1) showed that the time from the onset of COVID-19 to hospitalization is an important factor associated with the COVID-19 severity (AUC, 0.77 [95%CI, 0.69–0.84]; sensitivity, 0.73 [95%CI, 0.59–0.84]; specificity, 0.70 [0.59–

**Table 3. Clinical data in severity classification (n = 141).**

| | | Moderate group (n = 91) | Severe group (n = 50) | p value |
|---|---|---|---|---|
| Sex (Male, %) | | 55 (60.4) | 44 (88.0) | ***<0.001 |
| Age (y.o. Median ±SD, range) | | 59±17.5 (18–93) | 57±13.3 (30–93) | 0.312 |
| Period from onset to hospitalization (days, Median ±SD, range) | | 3±3.6 (0–13) | 8±4.2(0–22) | ***<0.001 |
| Period from onset to PCR positive (days, Median ±SD, range) | | 1±2.3 (0–13) | 2±2.3 (0–10) | 0.062 |
| Symptom (cases, %) | | | | |
| | fever | 79 (86.8) | 47 (94.0) | 0.192 |
| | dyspnea | 39 (42.9) | 40 (80.0) | ***<0.001 |
| | cough | 36 (39.6) | 18 (36.0) | 0.681 |
| | fatigue | 36 (39.6) | 12 (24.0) | 0.062 |
| | sore throat | 10 (11.0) | 0 (0) | 0.151 |
| | consciousness disorder | 1 (1.1) | 0 (0) | 0.462 |
| | headache | 4 (4.4) | 0 (0) | 0.132 |
| Height (cm, median ±SD, range) | | 162.5±10.8 (123–189) | 168.5±8.0 (148–181) | **0.005 |
| Weight (kg, median ±SD, range) | | 59.9±16.0 (34.1–121) | 74.1±16.5 (43.5–110) | ***<0.001 |
| BMI (median ±SD, range) | | 22.8±10.8 (13.8–45.5) | 26.3±8.0 (16.6–38) | ***<0.001 |
| smoking habit (cases, %) | | 42 (46.2) | 35 (70.0) | **0.007 |
| vaccination (cases, %) | | 44 (48.4) | 11 (22.0) | **0.002 |
| comorbidities (cases, %) | | | | |
| | respiratory disease | 14 (15.4) | 6 (12.0) | 0.584 |
| | cardiovascular disease | 23 (25.3) | 6 (12.0) | 0.062 |
| | renal disease | 32 (35.2) | 8 (16.0) | *0.016 |
| | continuous hemodialysis | 24 (26.4) | 6 (12.0) | *0.046 |
| | diabetes | 26 (28.6) | 12 (24.0) | 0.562 |
| | hypertension | 38 (41.8) | 23 (46.0) | 0.631 |
| | hyperlipidemia | 12 (13.2) | 13 (26.0) | 0.062 |
| | collagen diseases | 4 (4.4) | 1 (2.0) | 0.461 |
| | with malignant tumor | 8 (8.8) | 1 (2.0) | 0.112 |
| | pregnancy | 1 (1.1) | 1 (2.0) | 0.672 |
| | immunosuppression drugs | 5 (.5) | 2 (4.0) | 0.723 |
| Laboratory Data | | | | |
| | IL-6 (pg/ml) | 52.3±121.7 (0.2–646) | 31.3±75.3 (2.1–455) | 0.162 |
| | WBC ($10^3$/μl) | 6.1±3.3 (1.5–18.7) | 6.3±6.4 (2.7–45.5) | 0.871 |
| | CRP (mg/dl) | 5.7±6.8 (0.2–34.5) | 6.4±7.5 (0.1–37.8) | 0.221 |
| | D-dimer (μg/ml) | 1.2±13.2 (0.5–122) | 1.1±9.6 (0.5–67.3) | 0.852 |
| | AST (IU/l) | 34±40.2 (5–256) | 50.5±46.9 (21–252) | ***<0.001 |
| | ALT (IU/l) | 21±30.0 (3–165) | 50.5±41.3 (3–162) | ***<0.001 |
| | LDH (IU/l) | 300±136.7 (143–832) | 499.5±192.1 (224–1130) | ***<0.001 |
| | BUN (mg/dl) | 18±20.5 (1–91) | 19.5±19.2 (6–80) | 0.516 |
| | Cre (mg/dl) | 1.1±3.6 (0.4–12.7) | 0.89±3.9 (0.4–14.3) | 0.062 |
| | T.bil (mg/dl) | 0.5±0.3 (0.2–1.7 | 0.6±0.3 (0.3–1.8) | 0.162 |
| Period of whole hospitalization (days, Median ±SD, range) | | 9±5.4 (2–36) | 16±25.6 (2–159) | ***<0.001 |

(*Continued*)

**Table 3.** (Continued)

| | Moderate group (n = 91) | Severe group (n = 50) | p value |
|---|---|---|---|
| Outcome (Death, cases, %) | 1 (1.1) | 4 (8.0) | *0.032 |

y.o: year-old, SD: Standard Deviation, BMI: Body Mass Index, IL-6: Interleukin-6, WBC: White Blood Cell, CRP: C-reactive Protein, AST: Aspartate Aminotransferase, ALT: Alanine Aminotransferase, LDH: Lactate Dehydrogenase, Cre: Creatinine, T.bil: Total Bilirubin, Statistically significant difference

*$p<0.05$

**$p<0.01$

***$p<0.001$.

0.79]). The ROC also showed that the cut-off value for the time from the onset of COVID-19 to hospitalization was four days.

## Discussion

In this single-center retrospective study, we analyzed the characteristics, clinical features, and outcomes of COVID-19 caused by the delta and omicron variants of SARS-CoV-2 according to disease severity. The authors of previous reports [6, 7] have suggested that the hyperactivation of the inflammatory cascade, leading to a cytokine storm, is a critical biological response in patients with severe COVID-19. In addition, D-dimer, LDH, and CRP levels are reported to be potential biomarkers for COVID-19 severity [6, 7]. Significantly elevated levels of inflammatory cytokines TNF-α, IL-1, IL-6, and IL-10 have been documented in cases of severe COVID-19 compared to cases of non-severe disease [8–12]. In the present study, we did not

**Table 4. Therapeutic strategy between moderate group and severe group.**

| | | Moderate Group (n = 91) | Severe Group (n = 50) | p value |
|---|---|---|---|---|
| Oxygen supply (cases, %) | | 64 (70.3) | 50 (100) | ***<0.001 |
| Period of oxygenation (days, Median ±SD, range) | | 4.0±6.8 (0–43) | 15±26.2 (4–164) | ***<0.001 |
| Mechanical ventilation (cases, %) | | 0 (0) | 30 (60) | ***<0.001 |
| Period of ventilation (days, Median ±SD, range) | | 0(0) | 7.5±24.6 (3–133) | ***<0.001 |
| Tracheostomy (cases, %) | | 0 (0) | 7 (14) | ***<0.001 |
| ECMO (cases, %) | | 0 (0) | 5 (10) | ***<0.001 |
| NHF (cases, %) | | 3 (2.1) | 23 (46) | ***<0.001 |
| Period of NHF (days, Median ±SD, range) | | 9±3.5 (5–12) | 6±2.3 (1–10) | 0.112 |
| Prone position (cases, %) | | 3 (2.1) | 21 (42) | ***<0.001 |
| Period of prone position (days, Median ±SD, range) | | 4±1.5 (2–5) | 4±2.8 (2–12) | 0.681 |
| Therapy | | | | |
| | steroid | 69 (48.9) | 50 (100) | ***<0.001 |
| | remdesivir | 69 (48.9) | 50 (100) | ***<0.001 |
| | tocilizumab | 0 (0) | 3 (6) | *0.021 |
| | baricitinib | 3 (2.1) | 4 (8) | 0.222 |
| | casirivimab | 2 (1.4) | 0 | 0.291 |
| | molnupiravir | 16 (11.3) | 0 | **0.002 |
| | heparin | 61 (43.) | 50 (100) | ***<0.001 |

SD: Standard Deviation, ECMO: Extracorporeal Membrane Oxygenation, NHF: Nasal High Flow, Statistically significant difference

*$p<0.05$

**$p<0.01$

***$p<0.001$

**Table 5. Multivariable logistic regression (n = 141).**

|  | OR | 95% CI | *p*-value |
|---|---|---|---|
| The period between from onset to hospitalization | 1.16 | (1.02–1.32) | 0.026 |
| BMI | 1.10 | (1.01–1.21) | 0.039 |
| Smoking habit | 3.70 | (1.41–9.68) | 0.008 |
| LDH | 1.01 | (1.00–1.01) | < 0.001 |

OR: odds ratio; CI: confidence intervals, BMI: Body Mass Index, LDH: Lactate Dehydrogenase

focus on the levels of inflammatory cytokines for the classification of COVID-19 severity. The present results showed that LDH, which is reported to be a predictive biomarker for the aggravation of COVID-19 caused by the original variant of SARS-CoV-2, is involved in the aggravation of COVID-19 caused by the delta and omicron variants. Abnormalities in the levels of markers of cellular injury, particularly elevated LDH level, have been linked to greater disease severity [13, 14]. Data from recent studies suggest that LDH may be related to respiratory function and could be an important predictor of respiratory failure in patients with COVID-19 [15]. On the contrary, the present results indicated that IL-6, which has received considerable attention as a biomarker of the progression of COVID-19, is not a predictive factor for the aggravation of COVID-19. It has been reported [16–18] that SARS-CoV-2 infection increases the level of the inflammatory cytokine IL-6, which is believed to cause multiple organ damage. Therefore, administration of an IL-6 inhibitor has been established as a therapeutic strategy for COVID-19. However, as the IL-6 levels of the patients in the present study were not significantly high, even in the severe group, it could be concluded that IL-6 inhibitors may not be effective against COVID-19. The possible reasons for the lack of significant increase in IL-6 levels in the severe group are as follows: 1) the delta and omicron variants of SARS-CoV-2 could be less toxic than the original variant due to mutations, 2) SARS-CoV-2 may have mutated to induce inflammatory cytokines other than IL-6, and 3) IL-6 may not completely respond to SARS-CoV-2 infection due to the effect of vaccination. However, since vaccination was not yet widespread during the delta period, its effect may not have been extensive. In addition, since IL-6 level was measured on the first day of hospitalization in this study, it is unlikely that it was affected by the therapeutic drug administered. Thus, although we considered that various factors may be responsible for the lack of significant increase in IL-6 levels, the influence of the mutation of SARS-CoV-2 may be large. Furthermore, the lack of a significant increase in CRP and D-dimer levels was also considered to be due to the mutation of the virus.

It has been reported that cardiovascular disease, chronic renal disease, chronic lung diseases, diabetes mellitus, hypertension, immunosuppression, obesity, malignant tumor, and sickle cell disease are prognostic factors of COVID-19 severity [13, 19–22]. We investigated these comorbidities in the present study and found that except for obesity and renal disease, there were no significant differences between the comorbidities of the patients in the moderate and severe COVID-19 groups. However, significantly more patients with moderate COVID-19 had chronic renal failure and continuous hemodialysis than those with severe COVID-19. The reason for this finding is that it was difficult for patients to receive medical treatment in the outpatient dialysis clinic because chronic renal failure is one of the risk factors for aggravation of COVID-19. Regarding obesity, BMI >30, which is a proxy for obesity, was considered a strong predictor in a previous report [22]. There was a significant difference in BMI between the moderate and severe groups in the present study. During the delta period, patients with a high BMI developed more severe COVID-19. In addition, it was extremely difficult to manage their systemic conditions using mechanical ventilation, prone position therapy, and ECMO.

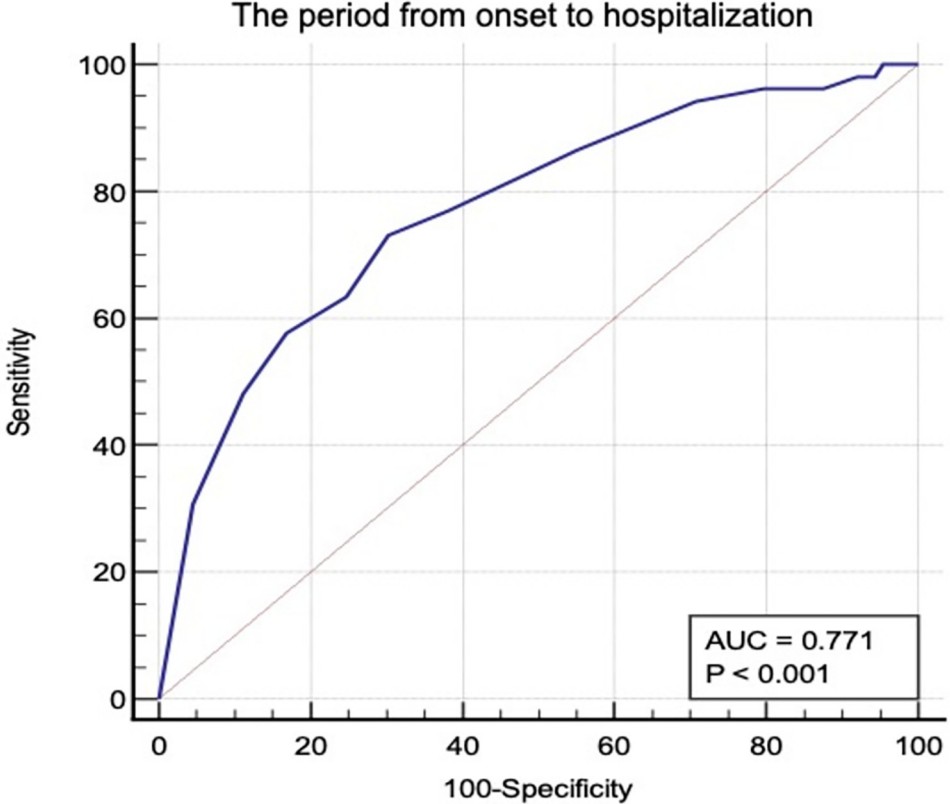

**Fig 1. Receiver Operatorating Characteristic of the period from onset to hospitalization related to severity of COVID-19 with area under the curve 0.77, sensitivity 0.73, specificity 0.70. From the ROC, the cut-off value of the period from onset of COVID-19 to hospitalization was 4 days.**

However, BMI was considered to be slightly associated with COVID-19 severity during the omicron period. It has been suggested that the effect of the COVID-19 vaccine may contribute to the prevention of aggravation in patients with a high BMI.

In this study, we analyzed diabetes mellitus and glycated hemoglobin level, which have been reported to be linked to inflammation, hypercoagulation, and high mortality (27.7%) [23], and found no significant differences between the moderate and severe groups. Therefore, abnormal glucose tolerance was not a predictor of aggravation in this study.

Regarding the time from the onset of COVID-19 to hospitalization, most patients felt some symptoms of COVID-19, such as fever, sore throat, and fatigue, and rested while waiting for recovery of their physical condition. In such a scenario, the respiratory status of the patient gradually worsens and the disease severity progresses. This is because after SARS-CoV-2 infection is established, the patient's respiratory condition gradually deteriorates and the patient is placed in a condition in which breathing is not difficult, even in a hypoxic state. This condition is called 'happy hypoxia' and is considered to be a precursor to the deterioration of respiratory function. This respiratory disorder caused by excessive spontaneous breathing without ventilator management is called patient self-inflated lung injury (P-SILI). This concept was previously described [24, 25] as a possible interplay between ventilation, surfactant dysfunction, and atelectasis during spontaneous ventilation, which leads to ventilation-induced lung injury. In clinical settings, intravenous injection of small doses of endotoxin in humans has a strong effect on respiratory drive, independent of fever or symptoms [26]. In addition, some patients without any pre-existing lung injury develop lung injury associated with hyperventilation [27].

Thus, in some patients, lung injury due to increased tidal volumes and ventilation may occur during spontaneous breathing, initiated by a high respiratory drive, which, in turn, leads to the development of lesions that appear similar to the ventilator-induced lung injury (VILI) observed in mechanically ventilated subjects. In these patients, the large spontaneous tidal volumes may be viewed as the cause of injury. Hence, any therapy that minimizes the generation of these large tidal volumes should be viewed as a prophylactic therapy against the progression of lung injury. Considering the above-mentioned concept, if a spontaneously breathing patient has a high respiratory drive that leads to increased minute ventilation with high tidal volumes, the goal of therapy must be to minimize P-SILI. If the patient engages in spontaneous breathing based on self-judgement, some parameters, such as changes in spontaneous breathing styles, tachypnea, changes in respiratory patterns, and oxygen demand, should be strictly managed, and any changes should be addressed immediately. If the changes are not managed immediately, the condition of the patient may become more severe. It is important to ascertain whether a spontaneously breathing patient has a high respiratory drive and has adopted a ventilatory pattern that will lead to subsequent lung injury. This concept is important for the management of respiratory conditions under mechanical ventilation for patients with severe COVID-19. The specific time from the onset of COVID-19 to aggravation and how long the deterioration of respiratory status should be observed have not been reported in any previous study. In the present study, the ROC showed that the optimal time from the onset of COVID-19 to hospitalization is four days. In other words, it is highly likely that COVID-19 will progress to severe if a patient's respiratory status is altered within four days after the onset of COVID-19, even if the patient is receiving home care; thus, immediate intervention is required. Proper intubation and a lung-protective ventilatory strategy guided by the severity of lung injury, including elevations in dead space, may be the easiest and most efficient way to achieve this goal. It is possible that by applying the same principles, mechanical ventilation can be administered "prophylactically or early" to protect the lung from P-SILI. As such, under defined conditions, mechanical ventilation, far from being just supportive or even damaging, becomes a true preventive measure against the progression of lung injury and perhaps ARDS. ECMO is another method that can be used to prevent P-SILI. When ventilator management for patients with severe COVID-19 reaches its limits, introduction of ECMO as a lung protection strategy may be expected for the prevention of or recovery from ARDS. However, regarding the introduction of ECMO, systemic management is necessary for weighing the possibility of the occurrence of serious complications caused by ECMO.

This study had several limitations. First, this was a single-center retrospective study performed in one of the leading hospitals in Japan, which has generally accepted and treated a large number of severely ill COVID-19 patients since the outbreak of disease. Unlike some countries, few hospitals in Japan collectively treat patients with COVID-19. Patients with COVID-19 are dispersed to nearby hospitals for treatment; thus, a single hospital does not treat many cases. Second, the moderate COVID-19 group in this study included elderly patients who did not need hospitalization for COVID-19 but could not stay alone at home due to lack of assistance from a caregiver. In addition, the moderate COVID-19 group included patients with mild COVID-19 who had several comorbidities and risk factors or patients who were hospitalized for the treatment of other illnesses but incidentally tested positive for COVID-19 without showing any symptoms. Thus, there may be some bias in our clinical data. Third, we did not establish exclusion criteria for this study and excluded patients undergoing treatment for other diseases to ensure that only the severity predictors of COVID-19 are considered. However, considering the spread of COVID-19 worldwide, the high infectivity of SARS-CoV-2, and the severity of patient symptoms, we analyzed the severity predictors without setting any exclusion criteria. Finally, this study focused on the identification of predictors

of the aggravation of COVID-19 using information available in routine practice, results of measurable specimen testing, and data on patient management based on standard of care. Combining these with the analysis of results of more advanced examinations and diagnostic imaging tests should further narrow down the severity predictors. However, although this may be possible for progressive research institutes, such as advanced medical institutions and universities, we believed that it was more important to distinguish COVID-19 severity in general medical institutions considering the spread of COVID-19 worldwide. The predictors identified this study, including the time from the onset of COVID-19 to hospitalization, LDH level at hospitalization, BMI, and smoking habits, are easy to understand and can be measured at any facility.

## Conclusion

This study showed that the time from the onset of COVID-19 to hospitalization is the most important factor in the prevention of the aggravation of COVID-19 caused by the delta and omicron variants of SARS-CoV-2. To prevent the aggravation of COVID-19, it is necessary to initiate appropriate medical management within four days after the onset of COVID-19, especially in patients with smoking habits, high BMI, and elevated LDH levels. These findings may facilitate preparations for the next wave of COVID-19 caused by other possible variants of SARS-CoV-2.

## Supporting information

**S1 Table. Patients' characteristics of different periods (Delta Period, Omicron Period) (n = 141).**
(DOCX)

**S2 Table. Patients' clinical features by severity classification of different periods (Delta Period & Omicron Period) (n = 141).**
(DOCX)

**S1 Fig. Relationship to severity and clinical and laboratory data represented by continuous variables.**
(TIF)

## Acknowledgments

We would like to thank our colleagues in Department of Emergency Medicine and clinical nurses in the intensive care unit of Yokohama City University Hospital for their kind assistance. And, we would like to thank Editage (www.editage.com) for English language editing.

## Author Contributions

**Conceptualization:** Fumihiro Ogawa.

**Data curation:** Fumihiro Ogawa, Takeru Abe.

**Formal analysis:** Fumihiro Ogawa, Takeru Abe.

**Investigation:** Fumihiro Ogawa, Yasufumi Oi, Hiroshi Honzawa, Naho Misawa, Tomoaki Takeda, Yushi Kikuchi, Ryosuke Fukui, Katsushi Tanaka, Daiki Kano, Hideaki Kato.

**Project administration:** Fumihiro Ogawa.

**Supervision:** Ichiro Takeuchi.

**Visualization:** Takeru Abe.

**Writing – original draft:** Fumihiro Ogawa.

**Writing – review & editing:** Fumihiro Ogawa.

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
