## [Decision Letter · Decision Letter 0]

22 Aug 2022

PONE-D-22-21582Severity Predictors of COVID-19 in SARS-CoV-2 Variant, Delta and Omicron Period; Single Center StudyPLOS ONE

Dear Dr. Ogawa,

Thank you for submitting your manuscript to PLOS ONE. After careful consideration, we feel that it has merit but does not fully meet PLOS ONE’s publication criteria as it currently stands. Therefore, we invite you to submit a revised version of the manuscript that addresses the points raised during the review process.

We look forward to receiving your revised manuscript.

Kind regards,

Benjamin M. Liu, MD, PhD

Academic Editor

PLOS ONE

Journal Requirements:

"No"

"No"

5. Please amend your manuscript to include your abstract after the title page.

Reviewers' comments:

Reviewer's Responses to Questions

**Comments to the Author**

1. Is the manuscript technically sound, and do the data support the conclusions?

Reviewer #1: Yes

Reviewer #2: Yes

2. Has the statistical analysis been performed appropriately and rigorously? 

Reviewer #1: Yes

Reviewer #2: Yes

3. Have the authors made all data underlying the findings in their manuscript fully available?

Reviewer #1: No

Reviewer #2: Yes

4. Is the manuscript presented in an intelligible fashion and written in standard English?

Reviewer #1: Yes

Reviewer #2: Yes

5. Review Comments to the Author

Reviewer #1: This article overcame the limitations of the clinical real world, finally collected 141 patients to explore the predictors of COVID-19 severity caused by SARS-CoV-2 variants. Research focused on the specific time from the onset of COVID-19 to aggravation, creatively followed with interest of 4 days of attention from onset to hospitalization of the patients, and suggested appropriate medical management within this time, which might provide reference for clinicians and future diagnosis and treatment.

In this paper, the characteristics and data of 141 enrolled patients were analyzed statistically, which provided support for the above conclusions. However, there are still some problems, which might be troubled the support.

1.Table 5

We can see that the four indicators have statistical significance. We know that you must have done the ROC curve analysis for the other three indicators, and in order to highlight the key points. The problem is that it seems that only four positive indicators are selected and the findings are highlighted. After all, the OR value of Smoking is the largest among the four indicators, and the p value of LDH is the largest among the four indicators. Why the period is the final target ? And it is shown as a unique figure.

Looking at the above question from another view, it is suggested in Table 3 that Gender, Dyspnea, Height and other indicators showed significant differences.

In addition, Figure 2 of reference 12 provides a good data display of the ROC curve analysis.

2.Table 5

From the results in Table 5, which is the most important argument supporting the conclusion, there seems to be no separate statistics for Delta and Omicron, but regression analysis for disease severity. It is not difficult to understand the painstaking efforts of clinical observation to divide the virus variants into two time periods, however, it raises a new question that are all 89 enrolled patients diagnosed with delta typing in the delta time period, and all 52 omicron?

This prompts us to pay attention to the contents in supplementary Table 1. It seems that only part of contents in supplementary Table 2 is involved in supplementary Table 1. The data of clinical observation, treatment and laboratory diagnosis are missing or selectively deleted. This part of data can be used as a reference for readers to understand the comparison between Delta and Omicron period. It is suggested to supplement this part of data. And the supplementary Tables 1 and 2 might be displayed together. After all, supplementary Table 2 is a stratification reanalysis of the data in supplementary Table 1, according to disease severity.

Further more, as discussed in the article Line 189-192, the impact of mutations may be huge, so why not analyze and display the existing data? Line144-145, what are the factors between the two periods?

3.Table 2

The same data missing display occurred in Table 2. Compared with Table 3, it can be seen that there is a lack of baseline data of laboratory diagnosis and the length of hospitalization which is the most relevant data to the focus of the paper.

It is suggested that Table 2 and Table 3 can be combined. Although it could be understandable that Table 2 needs to be more concise, please refer to the data display mode of Table 1 in reference 14, which may provide more information. After all, the data in Table 3 is also the continuation of the stratification reanalysis of the data in Table 2.

The above three questions are the main questions of the manuscript and the places that need to be analyzed. Next questions 4-15, there are some specific details that need to pay attention, but no longer affect the logic of the full text.

4. Numerical standard of SpO2

There are three numerical value of 93% in Line 45, 94% in Line 46 and 92% in Table 1.

In particular, the first two values may be confusing to readers, because they are very close.

5.Median age of 141 patients

59.6 in Line 95, meanwhile 58 in Table 2.

6.Range of age of moderate group

18-83 in Table 3, meanwhile 19-93 in Line 112.

7.Cases of vaccination

According to supplementary Table 2, the two p values should be 0.165 and 0.595, respectively, with the latter one missing.

47 and 39 in Table 3 seem to represent the number of patients without immunization, which should be 44 and 11, respectively.

Can these data support the conclusion in Line 105-106?

8.Cases of comorbidities

23 in 50 (46%) of patients in severe group have hypertension, although no significant discussion in Line 125.

9.Table 4

Baricitinib and casirivimab have no significance, while tocilizumab and molnupiravir include the case where the admitted patient is 0. However, discussion in Line 139 says each treatment.

10.Line 167

We did not focus, however, a whole section from Line 168 to 192 was discussed.

11.Line 196

There were no significances except obesity, however, renal disease and continuous hemodialysis were in Table 3, although it explained in the following paragraph. And additional references are required in Line 201 to help readers understanding, instead of listing them in Line 195.

12.Line 212

The negative data conclusion is not shown in the chart. This is not a problem. The key point is part of the data of the full text seems to consider displaying in a scatter chart instead of a full table, as Figure 1 in reference 23.

In addition, Figure 1 in reference 15 and Figure 2 in reference 23 are also another good way to display data of correlation analysis.

13.Line 247

Previous studies of supplementary Table 1 in correction version of reference 10 compared time from onset of symptom to test of two groups, so do Table 1 in reference 14, Table 1 in reference 19, and Table 1 in reference 20.

14.Line 251-253

Is the ultimate reason, that the treatment plan of the one unfortunate patient in the moderate group in Table 3, just violates this way mentioned here?

15.P value

For the p value in the full text, especially in the table, please uniformly keep the digits after the decimal point, and it is better to mark the p value of statistical significance with an asterisk or other common symbols.

Reviewer #2: In this study, a total of 141 patients were enrolled and divided into two groups according to disease severity to analyze the characteristics, clinical features, and outcomes of COVID-19 caused by the delta and omicron variants of SARS-CoV-2. The article has unique significance. I appreciate the work, however if the pulmonary CT of cases can be supplemented, the article will be more complete and full.

6. PLOS authors have the option to publish the peer review history of their article (what does this mean?). If published, this will include your full peer review and any attached files.

Reviewer #1: No

Reviewer #2: No

---

## [Author Response · Author response to Decision Letter 0]

8 Sep 2022

Response to Reviewer #1

Comment of Reviewer #1

1.Table 5

We can see that the four indicators have statistical significance. We know that you must have done the ROC curve analysis for the other three indicators, and in order to highlight the key points. The problem is that it seems that only four positive indicators are selected and the findings are highlighted. After all, the OR value of Smoking is the largest among the four indicators, and the p value of LDH is the largest among the four indicators. Why the period is the final target? And it is shown as a unique figure.

Looking at the above question from another view, it is suggested in Table 3 that Gender, Dyspnea, Height and other indicators showed significant differences.

In addition, Figure 2 of reference 12 provides a good data display of the ROC curve analysis.

Thank you for your comment for the result of Table 5. We agreed with you that the OR value of Smoking is the largest among the four indicators, and the p value of LDH is the largest among the four indicators. These factors are very important to predict severity of COVID-19 pneumonia compared to the period from onset to hospitalization. We focused the period from onset to hospitalization in this study because no one knows it specifically in previous report. In this study, by clarifying the specific cut-off value of the period from onset to hospitalization, we considered the possibility that it could be a predictor. Then, we added two sentences of this point in Line 16-17 and Line 23-24 in Revised Manuscript with Tracking Change.

2.Table 5

From the results in Table 5, which is the most important argument supporting the conclusion, there seems to be no separate statistics for Delta and Omicron, but regression analysis for disease severity. It is not difficult to understand the painstaking efforts of clinical observation to divide the virus variants into two time periods, however, it raises a new question that are all 89 enrolled patients diagnosed with delta typing in the delta time period, and all 52 omicron?

This prompts us to pay attention to the contents in supplementary Table 1. It seems that only part of contents in supplementary Table 2 is involved in supplementary Table 1. The data of clinical observation, treatment and laboratory diagnosis are missing or selectively deleted. This part of data can be used as a reference for readers to understand the comparison between Delta and Omicron period. It is suggested to supplement this part of data. And the supplementary Tables 1 and 2 might be displayed together. After all, supplementary Table 2 is a stratification reanalysis of the data in supplementary Table 1, according to disease severity.

Furthermore, as discussed in the article Line 189-192, the impact of mutations may be huge, so why not analyze and display the existing data? Line144-145, what are the factors between the two periods?

Thank you for your comment. We are very sympathetic to the points you focused. In fact, the current study only separates the mutant strains at the period when they were considered to be the majority of the strains, which did not mean that the strains are accurately identified in the PCR test. If we had been able to identify all of these strains with certainty, we would have been able to present the characteristics in each strain in more detail. For this reason, we presented this as a supplement table because there was no certainty of the characteristics of the delta and omicron variants in the main text

As for the supplement table2, all the data were presented and we don’t think there are any missing parts, but is there anything missing? We described all of data in all case without any selection bias.

3.Table 2

The same data missing display occurred in Table 2. Compared with Table 3, it can be seen that there is a lack of baseline data of laboratory diagnosis and the length of hospitalization which is the most relevant data to the focus of the paper.

It is suggested that Table 2 and Table 3 can be combined. Although it could be understandable that Table 2 needs to be more concise, please refer to the data display mode of Table 1 in reference 14, which may provide more information. After all, the data in Table 3 is also the continuation of the stratification reanalysis of the data in Table 2.

Thank you for your comment. We have carefully reviewed the missing data that you focused, and we have not been able to find any missing data, and we have included all of the laboratory data, length of hospital stay, and so on. 

Regarding the integration of table 2 and table 3, we thought it would be easier to understand if we described the overall characteristics of all patients as a whole (table 2), and then separate moderate and severe COVID-19 cases and compared further clinical and laboratory data. So, we have separated table 2 and table 3. 

4. Numerical standard of SpO2

There are three numerical value of 93% in Line 45, 94% in Line 46 and 92% in Table 1.

In particular, the first two values may be confusing to readers, because they are very close.

Thank you for your comment for confusing you about the standard value of SpO2 for severe COVID-19. We made a mistake for numerical standard of SpO2. We consolidated the standard value of SpO2 according to global standard.

5.Median age of 141 patients

59.6 in Line 95, meanwhile 58 in Table 2.

Thank you for your comment. I made a mistake of median age of 141 patients. 58 was correct value of median age. We corrected it.

6.Range of age of moderate group

18-83 in Table 3, meanwhile 19-93 in Line 112.

Thank you for your comment. I made a mistake of range of age of moderate group in manuscript. The value in Table 3 was correct. So, we corrected it.

7.Cases of vaccination

According to supplementary Table 2, the two p values should be 0.165 and 0.595, respectively, with the latter one missing.

47 and 39 in Table 3 seem to represent the number of patients without immunization, which should be 44 and 11, respectively.

Can these data support the conclusion in Line 105-106?

Thank you for your comment. You are right. I made a mistake of the value of vaccination. I corrected this value. The data support the conclusion in Line 109-110.

8.Cases of comorbidities

23 in 50 (46%) of patients in severe group have hypertension, although no significant discussion in Line 125.

Thank you for your comment. We already described “No significant frequency of any comorbidity was observed in the severe group” in Line 141 of Revised Manuscript with Tracking Change.

9.Table 4

Baricitinib and casirivimab have no significance, while tocilizumab and molnupiravir include the case where the admitted patient is 0. However, discussion in Line 139 says each treatment.

Thank you for your comment for the therapeutic strategy. You are correct. Casirivimab and molnupiravir were only used for moderate cases of COVID-19 and not for severe cases, so there were significant differences There was a significant difference from therapeutic strategy in Table 1 because remdisivir and steroid were absolutely used for severe cases of COVID-19. We replaced these sentences to correct sentences in Line 156-158 in Revised Manuscript with Tracking Change. “There were differences in treatment between the two groups in terms of the characteristics of each treatment due to differences in treatment between moderate and severe cases of COVID-19 as indicated by therapeutic strategy in table 1.”

10.Line 167

We did not focus, however, a whole section from Line 168 to 192 was discussed.

Thank you for your comment. The purpose of this sentence was that many of the biochemical factors that have been reported as existing severity predictors were in the early stages of COVID-19, and most of them have been reported as severity predictors after the start of vaccination and standard therapy for COVID-19. Since it had not been done, I decided to discuss this here because I paid attention to whether these severity predictors changed after vaccination and treatment methods were standardized. Originally, LDH, CRP, D-dimer, and IL-6 were listed as severity predictors of COVID-19, but in our study, only LDH was listed as a significant severity predictor. We thought that this could be done because it was transformed due to vaccinations and standard treatment for COVID-19.

11.Line 196

There were no significances except obesity, however, renal disease and continuous hemodialysis were in Table 3, although it explained in the following paragraph. And additional references are required in Line 201 to help readers understanding, instead of listing them in Line 195.

Thank you for your comment. As you said, we thought this paragraph was confusing to the readers, so we changed the sentence in Line 220 and 223. As written in the main text, COVID-19 patients with renal failure and hemodialysis had high risk of severity, so they cannot be managed on an outpatient clinic. Due to this reason, mild COVID-19 patients with chronic renal disease and hemodialysis tended to hospitalize more often. 

12.Line 212

The negative data conclusion is not shown in the chart. This is not a problem. The key point is part of the data of the full text seems to consider displaying in a scatter chart instead of a full table, as Figure 1 in reference 23.

In addition, Figure 1 in reference 15 and Figure 2 in reference 23 are also another good way to display data of correlation analysis.

Thank you for your thoughtful comment. Accordingly, we added figures including negative data posted a box plot for the relationship between severity of COVID-19 and clinical data represented by continuous variables. We put them in the supplementary material (Supple Fig.1), so that we would be able to avoid redundancy of information in the main manuscript (Line 150-152). 

13.Line 247

Previous studies of supplementary Table 1 in correction version of reference 10 compared time from onset of symptom to test of two groups, so do Table 1 in reference 14, Table 1 in reference 19, and Table 1 in reference 20.

Thank you for your comment. We knew the data for correlation with period from illness onset to hospital admission in these reports. These data obtained from the conventional strain data of SARS-CoV-2. In clinical practice, the incubation period, symptom onset period, and period of transition to severe COVID-19 in variants of SARS-CoV-2 seemed to be shortened compared to conventional strains, so we focused on this point again. As mentioned in the discussion, as the factors for the establishment of COVID-19 pneumonia are being gradually elucidated, it was important to consider that the concept of P-SILI can occur during the waiting period for medical treatment. This is the reason of this sentence.

14.Line 251-253

Is the ultimate reason, that the treatment plan of the one unfortunate patient in the moderate group in Table 3, just violates this way mentioned here?

Thank you for your comment. The only patient who died of moderate disease was an elderly patient with renal failure and hemodialysis who unfortunately died of natural causes because he did not wish to undergo further invasive treatment, in spite of severe progression.

15.P value

For the p value in the full text, especially in the table, please uniformly keep the digits after the decimal point, and it is better to mark the p value of statistical significance with an asterisk or other common symbols.

Thank you for your comment. We agreed with you and amended the consistent decimal for p-values with asterisk for significant differences in the revised manuscript. 

Response to Reviewer #2

Comment of Reviewer #2: In this study, a total of 141 patients were enrolled and divided into two groups according to disease severity to analyze the characteristics, clinical features, and outcomes of COVID-19 caused by the delta and omicron variants of SARS-CoV-2. The article has unique significance. I appreciate the work, however if the pulmonary CT of cases can be supplemented, the article will be more complete and full.

Thank you for your comment. We appreciate your interest in our work. We agreed with your comment that the pulmonary CT of these cases are more interesting to evaluate severity of COVID-19. Other researcher evaluates the CT score of these cases from chest CT scan for another article now, so we can’t use the data for this article. We apologize you.

---

## [Decision Letter · Decision Letter 1]

10 Oct 2022

Severity Predictors of COVID-19 in SARS-CoV-2 Variant, Delta and Omicron Period; Single Center Study

PONE-D-22-21582R1

Dear Dr. Ogawa,

We’re pleased to inform you that your manuscript has been judged scientifically suitable for publication and will be formally accepted for publication once it meets all outstanding technical requirements.

Kind regards,

Benjamin M. Liu, MD, PhD

Academic Editor

PLOS ONE

Additional Editor Comments (optional):

Reviewers' comments:

Reviewer's Responses to Questions

**Comments to the Author**

1. If the authors have adequately addressed your comments raised in a previous round of review and you feel that this manuscript is now acceptable for publication, you may indicate that here to bypass the “Comments to the Author” section, enter your conflict of interest statement in the “Confidential to Editor” section, and submit your "Accept" recommendation.

Reviewer #1: All comments have been addressed

Reviewer #2: All comments have been addressed

Reviewer #3: All comments have been addressed

2. Is the manuscript technically sound, and do the data support the conclusions?

Reviewer #1: Yes

Reviewer #2: Yes

Reviewer #3: Yes

3. Has the statistical analysis been performed appropriately and rigorously? 

Reviewer #1: Yes

Reviewer #2: Yes

Reviewer #3: Yes

4. Have the authors made all data underlying the findings in their manuscript fully available?

Reviewer #1: Yes

Reviewer #2: Yes

Reviewer #3: Yes

5. Is the manuscript presented in an intelligible fashion and written in standard English?

Reviewer #1: Yes

Reviewer #2: Yes

Reviewer #3: Yes

6. Review Comments to the Author

Reviewer #1: The revise manuscript explores the predictors of COVID-19 severity caused by SARS-CoV-2 variants in 141 enrolled patients, and focused on the specific time from the onset of COVID-19 to hospitalization, which might provide reference for clinicians and future diagnosis and treatment. The authors can be able to review opinions one by one and make modifications or explanations. I appreciate the research in this article and recommend.

But I must also point out that, compared with Table 3, the data in Table 2 only includes data from sex to comorbidities in list column, missing from laboratory data to outcome. Similar situations occur in Supplementary Table 1, compared with Supplementary Table 2. However, the method of presentation of the data does not affect the current logic of the paper. I also perceive the author's explanation.

Reviewer #2: (No Response)

Reviewer #3: Although this paper is a retrospective, single-center study, the data provided are relatively detailed and the statistical methods are appropriate, which can support the conclusions drawn by the study. I personally think this paper can be published.

7. PLOS authors have the option to publish the peer review history of their article (what does this mean?). If published, this will include your full peer review and any attached files.

Reviewer #1: No

Reviewer #2: No

Reviewer #3: **Yes: **Aimei Liu

---

## [Editor Report · Acceptance letter]

14 Oct 2022

PONE-D-22-21582R1 

Severity Predictors of COVID-19 in SARS-CoV-2 Variant, Delta and Omicron Period; Single Center Study 

Dear Dr. Ogawa:

I'm pleased to inform you that your manuscript has been deemed suitable for publication in PLOS ONE. Congratulations! Your manuscript is now with our production department. 

Kind regards, 

on behalf of

Dr. Benjamin M. Liu 

Academic Editor

PLOS ONE